# DIFFUSION WITH TRUNCATED BLOCKS: FAST AND HIGH-QUALITY TEXT GENERATION USING TRUNCATED BLOCK GENERATION

## ABSTRACT

Diffusion-based Large Language Models (dLLMs) are emerging as a powerful alternative to traditional autoregressive models. These models learn to generate text by iteratively denoising masked sequences. In this work, we identify a critical problem in dLLMs that using token-level noise: the model's attention is wastefully expended on uninformative mask tokens, diluting its focus on meaningful context. We term this phenomenon "attention dilution". We further show that it is an artifact of token-level noising, whereas models with sequence-level noise does not have such phenomenon. To resolve this problem, we introduce Truncated Block Generation, a novel sampling algorithm that not only mitigates attention dilution but also enables faster inference and flexible-length sequence generation. Extensive experiments validate our analysis and demonstrate the marked effectiveness of our proposed method in enhancing both the performance and efficiency of dLLMs.

## 1 INTRODUCTION

Diffusion large language models (dLLMs) (Nie et al., 2025; Ye et al., 2025; Zhu et al., 2025; Khanna et al., 2025) have recently emerged as an alternative promising paradigm for language modeling. While autoregressive models (ARMs) (Achiam et al., 2023; Liu et al., 2024; Dubey et al., 2024) generate text token-by-token in a strict left-to-right manner, dLLMs operate on a sequence of masked tokens, iteratively refining the entire sequence in parallel. This non-autoregressive, denoising approach enables bidirectional attention, parallel generation and more flexible generation patterns, directly addressing some of the inherent limitations of ARMs.

The standard dLLM employs a uniform, sequence-level noise, where all masked tokens in a sequence have equal importance (Nie et al., 2025). As illustrated in Figure 1(a), as the same loss weight is assigned to every masked position, its surrounding context is ignored. A more nuanced approach is proposed in Dream (Ye et al., 2025), which uses token-level noise and dynamically re-weights the loss for each token based on its contextual informativeness. For example, as illustrated in Figure 1(b), a masked token surrounded by unmasked neighbors is considered more informative, and is thus assigned a higher loss weight. This encourages the model to prioritize predicting tokens with rich contextual support during inference, leading to a more structured generation process.

However, we identify a critical and previously overlooked drawback of token-level noise. We provide empirical and theoretical evidence that as the number of masked tokens in the generation context grows, the informativeness of each individual mask token decreases significantly. For models trained with token-level noise, attending to a long sequence of these low-information mask tokens dilutes the model's attention, preventing it from focusing on the truly informative tokens in the prompt and the partially informative mask tokens. This "attention dilution" degrades generation quality, particularly for long sequences. Conversely, our analysis reveals this is not an issue for models trained with sequence-level noise, where all mask tokens are trained to carry useful information and contribute meaningfully to the self-attention mechanism.

Inspired by this core insight, we propose Truncated Block Generation, a novel sampling method designed specifically to mitigate attention dilution in dLLMs trained with token-level noise. Instead of appending a single, long sequence of masks to the prompt, our method divides the generation process into sequential rounds. In each round, a shorter block of masks is appended and denoised, and

| w(t) = 1.0 | Diffusion | Large | Language | Models |
|---|---|---|---|---|

| w(t) = 0.5 | Diffusion | [MASK] | Language | [MASK] |
|---|---|---|---|---|

| w(t) = 0.0 | [MASK] | [MASK] | [MASK] | [MASK] |
|---|---|---|---|---|

| | Diffusion | [MASK] | Language | Models |
|---|---|---|---|---|
| | w(t) = 1.0 | w(t) = 0.62 | w(t) = 0 | w(t) = 0 |

| | Diffusion | Large | [MASK] | [MASK] |
|---|---|---|---|---|
| | w(t) = 1.0 | w(t) = 1.0 | w(t) = 0.37 | w(t) = 0.18 |

(a) Sequence-Level Noise.  (b) Token-Level Noise.

Figure 1: The noise level of masked diffusion language models: Different from LLaDA, which uses sequence-level noise, Dream uses token-level noise to re-weight the loss of different tokens.

we use the truncated unmasked block as the new context. This strategy ensures that the model always attends to a context with a high density of informative tokens, alleviating the dilution problem. As a result, Truncated Block Generation accelerates sampling speed, supports flexible-length generation, and significantly improves output quality.

Our main contributions are summarized below.

1. We identify the "attention dilution" problem in dLLMs trained with token-level noise. We demonstrate that an excessive number of mask tokens degrades performance by diverting the model's attention away from informative context.

2. We provide a theoretical analysis explaining the underlying mechanism of attention dilution. Furthermore, we clarify why models trained with sequence-level noise are not susceptible to this issue, highlighting a critical difference in their behavior.

3. Based on our analysis, we propose Truncated Block Generation, a simple yet effective sampling strategy that directly mitigates attention dilution. Extensive experiments validate that our method leads to substantial improvements in generation quality, faster inference speed, and robust support for flexible-length outputs.

## 2 BACKGROUND

### 2.1 DIFFUSION LARGE LANGUAGE MODELS

Discrete diffusion models (Sohl-Dickstein et al., 2015; Meng et al., 2022; Austin et al., 2021a) are a class of latent variable models of the form $p_{\boldsymbol{\theta}}(\boldsymbol{x}_0) = \int p_{\boldsymbol{\theta}}(\boldsymbol{x}_0 : T) \, \mathrm{d}\boldsymbol{x}_{1:T}$. They are characterized by a forward noising process and a learned reverse denoising process. The forward process progressively corrupts the original data $\boldsymbol{x}_0$ into a sequence of increasingly noisy masked tokens $\boldsymbol{x}_1, \ldots, \boldsymbol{x}_T$, which are of the same dimensionality as the data $\boldsymbol{x}_0 \sim p_{\text{data}}$: $q(\boldsymbol{x}_{1:T} \mid \boldsymbol{x}_0) = \prod_{t=1}^{T} q(\boldsymbol{x}_t \mid \boldsymbol{x}_{t-1})$, where $q(\boldsymbol{x}_t \mid \boldsymbol{x}_s) = \text{Cat}(\boldsymbol{x}_t; \mathbf{Q}_t \boldsymbol{x}_s)$ and $\mathbf{Q}_t$ is the transition matrix. The backward process learns to gradually denoise the masked sequence back to the original data distribution by iteratively predicting masked tokens as $t$ moves from $T$ to 0: $p_{\boldsymbol{\theta}}(\boldsymbol{x}) = \sum_{\boldsymbol{x}_{1:T}} p(\boldsymbol{x}_T) \prod_{t=1}^{T} p_{\boldsymbol{\theta}}(\boldsymbol{x}_{t-1} | \boldsymbol{x}_t)$.

The model parameter $\boldsymbol{\theta}$ can be optimized by minimizing the negative log-likelihood of the clean data $\boldsymbol{x}_0$. If the model uses the absorbing kernel, learning the denoising process leads to the minimization of the following Evidence Lower Bound (ELBO) of the log likelihood (Nie et al., 2025):

$$\mathcal{L}_{\text{LLaDA}}(\theta) = -\mathbb{E}_{\boldsymbol{x} \sim q(\boldsymbol{x}), \epsilon \sim \mathcal{N}(0,1), t \sim \mathcal{U}(1,T)} w(t) \sum_{n=1}^{N} \mathbf{1}_{[\boldsymbol{x}_t^n = \text{MASK}]} \log p_{\boldsymbol{\theta}}(\boldsymbol{x}_0^n \mid \boldsymbol{x}_t), \tag{1}$$

where $\mathbf{1}_{[\boldsymbol{x}_t^n = \text{MASK}]}$ is the indicator function that ensures that the loss is computed only on the masked tokens, and $w(t) \in (0, 1]$ is a time-dependent reweighting term. To enable the model to decode easier tokens earlier, Ye et al. (2025) propose using contextual token-level noise for token loss reweighting:

$$\mathcal{L}_{\text{Dream}}(\theta) = -\mathbb{E}_{\boldsymbol{x}_0 \sim q(\boldsymbol{x}_0), \epsilon \sim \mathcal{N}(0,1), t \sim \mathcal{U}(1,T)} \sum_{n=1}^{N} w(\boldsymbol{x}_t, t, n) \mathbf{1}_{[\boldsymbol{x}_t^n = \text{MASK}]} \log p_{\boldsymbol{\theta}}(\boldsymbol{x}_0^{n-1} \mid \boldsymbol{x}_t),$$

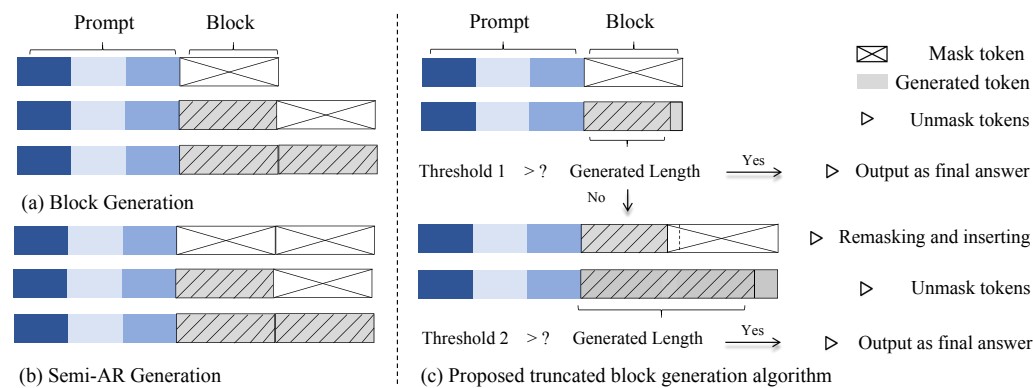

Figure 2: Comparison of block generation, semi-AR generation and our truncated block generation.

where

$$w(\boldsymbol{x}_t, t, n) = \frac{1}{2} \sum_{i=1}^{N} \mathbf{1}_{\left[\boldsymbol{x}_t^i \neq \texttt{MASK}\right]} \text{Geo}(p, |n - i| - 1) \tag{2}$$

is a mixture of geometric distributions to quantify the information contribution of each clean token relative to the noised tokens.

## 2.2 BI-DIRECTIONAL ATTENTION OF DIFFUSION LANGUAGE MODELS

For an input sentence $X \in \mathbb{R}^{1 \times l}$, the dLLMs first projects each token to its embedding $X_0 \in \mathbb{R}^{1 \times l \times d}$, where $d$ is the hidden size. Unlike AR transformers, which add an unidirectional causal attention mask to the attention score and generate text sequentially in a left-to-right manner, MDM's transformer uses bi-directional attention and attends to both the previous and future tokens.

$$Q_0 = X_0 W_Q, \quad K_0 = X_0 W_K, \quad A^0 = (\frac{Q_0 K_0^\top}{\sqrt{d}}), \quad \text{Attn}(X_0) = \text{Softmax}(A^0).$$

Thus, on inference, the MDM (i) attends to the previous prompt tokens to understand the question, and (ii) attends to the following [MASK] tokens for generating and organizing the content.

## 3 THE ATTENTION DILUTION PROBLEM IN MASKED DIFFUSION MODELS

MDMs generate sequences by appending a series of $m_1$ mask tokens $M_1 \in \mathbb{R}^{1 \times m_1}$ to the end of a prompt $P_0 \in \mathbb{R}^{1 \times c}$. The combined sequence is then projected into an embedding space $X_1 \in \mathbb{R}^{1 \times (c + m_1) \times d}$:

$$X_1 = \texttt{wte}\left(\begin{bmatrix} P_0 \\ M_1 \end{bmatrix}\right) \in \mathbb{R}^{1 \times (c + m_1) \times d}, \tag{3}$$

where $\texttt{wte}$ is the word token embedding layer. While intuitive, this introduces a critical challenge (which will be called *attention dilution*), particularly for models trained with token-level noise. For a mask token $\boldsymbol{x}_j$ in $X_1$, we found that with the increase of the token's positions, its contextual information

$$\frac{w(\boldsymbol{x}_{j+1}, j + 1, c + m_1)}{w(\boldsymbol{x}_j, j, c + m_1)} = \frac{\sum_{i=0}^{c+m_1-1} p^{|n-i+1|}}{\sum_{i=1}^{c+m_1} p^{|n-i|}} = p.$$

**Theorem 1.** *When the length of the mask-appended sentence is sufficiently long, for every $\epsilon > 0$, there exists $k > 0$ such that $w(\boldsymbol{x}_u, u, n) < \epsilon, \ \forall u > k$.*

For tokens with contextual information less than $\epsilon$, we call them no-informative tokens, and the model attending on these tokens can only get positional information (know the length of the remaining free space for generation), but this is not helpful for understanding and solving the problem. We define the attention on tokens whose contextual information is greater than $\epsilon$ as contextual attention:

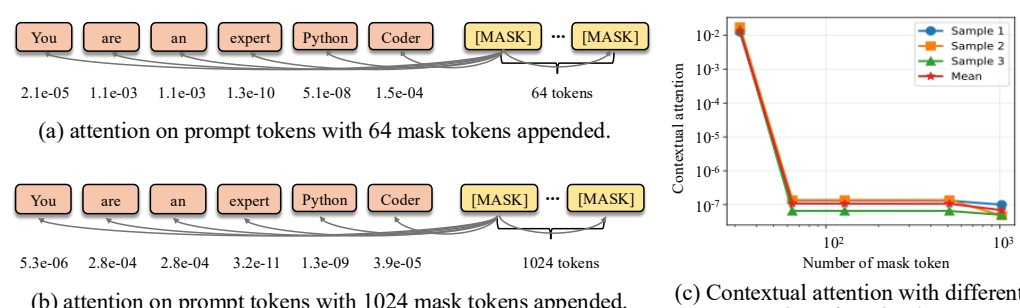

(a) attention on prompt tokens with 64 mask tokens appended.

(b) attention on prompt tokens with 1024 mask tokens appended.

(c) Contextual attention with different number of mask token.

Figure 3: The attention of Dream will be diluted by excessive masks.

**Definition 1.** *We denote the ith mask token of $X_1$ as $\boldsymbol{m}_i^1$. We define its contextual attention as its attention on tokens whose contextual information is greater than $\epsilon$:*

$$CA(\boldsymbol{m}_i^1) = \sum_{j=1}^{c+m_1} \mathbf{1}_{[w(x_j,j,n)>\epsilon]} Attn_{ij}(X_1).$$

Next, we prove that the excessive redundant mask tokens dilutes the contextual attention.

**Theorem 2.** *For the $X_1$ defined above, we further append $m_2 - m_1$ mask tokens to its end and denote it as $X_2$:*

$$X_1 = \mathtt{wte}\left(\begin{bmatrix} P_0 \\ M_1 \end{bmatrix}\right) \in \mathbb{R}^{1\times(c+m_1)\times d}, \quad X_2 = \mathtt{wte}\left(\begin{bmatrix} P_0 \\ M_2 \end{bmatrix}\right) \in \mathbb{R}^{1\times(c+m_2)\times d}. \quad (4)$$

*We denote the ith mask token of $X_1$ as $\boldsymbol{m}_i^1$ and the ith mask token of $X_2$ as $\boldsymbol{m}_i^2$. The contextual attention of $\boldsymbol{m}_i^1$ is strictly greater than the contextual attention of $\boldsymbol{m}_i^2$:*

$$CA(\boldsymbol{m}_i^1) = \sum_{j=1}^{c+m_1} \frac{\exp(A_{ij}^1)}{\sum_{k=1}^{c+m_1} \exp(A_{ik}^1)} > \sum_{j=1}^{c+m_1} \frac{\exp(A_{ij}^2)}{\sum_{k=1}^{c+m_2} \exp(A_{ik}^2)} = CA(\boldsymbol{m}_i^2) \quad (5)$$

We empirically validate this phenomenon in Figure 3. As we can see in Figure 3 (a) and Figure 3 (b), given a prompt "You are a expert python coder", when the number of appended [MASK] is 64, the attention of the first mask token to the [You] token is $2.1 \times 10^{-5}$. However, if the number of appended [MASK] is 1024, the attention of the first mask token to the [You] token will decrease to $5.3 \times 10^{-6}$. The same phenomenon can be observed for attention on all of the prompt tokens. Moreover, we also visualize how the contextual attention affected by the number of mask tokens and show the result in Figure 3 (c). We draw the curve of three samples and the mean on the MBPP dataset, and we can see that the contextual attention is decreasing with the increase of the number of mask tokens. This demonstrates that an excess of masks diverts focus from the crucial context provided by the prompt.

**Remarks.** The attention dilution problem is significant in MDMs trained with token-level noise but is absent in models trained with sequence-level noise.

Recall that in attention layer, $q^\top k$ measures the similarity of different tokens, so masks tokens will attend more to mask tokens rather than prompt tokens. However, prompt tokens are more important for the generation. We observe that for both Dream (Ye et al., 2025) and LLaDA (Nie et al., 2025), they will copy a previous token as query for self-attention in the following layers. As we can see in Figure 5 (a) and Figure 5 (b), for the mask tokens, their attention will be paid to one token in the prompt, we denote the index of the prompt token as $k$ and the mask token as $\boldsymbol{m}$:

$$\text{Attn}_{\boldsymbol{m}}(X_0) = \boldsymbol{e}_k, \quad \text{Attn}_{\boldsymbol{m}}(X_0)\boldsymbol{v} = \boldsymbol{e}_k^\top \cdot \left[X_0^1 W_V, \cdots X_0^k W_V, \cdots\right] = X_0^k W_V, \quad (6)$$

where $\boldsymbol{e}_k$ denotes the unit vector with a 1 in the k-th position and zeros elsewhere. The attention layer will output the linear transformed token $X_0^k W_V$ in all the mask tokens' positions and copy it to these positions using residual connection.

$$\boldsymbol{m} + \text{Layer}(X_0) = \boldsymbol{m} + X_0^k W_v \quad (7)$$

```
Prompt: Write a python function to find
the count of rotations of a binary
string with odd value.
[BEGIN]

First round generation:
def odd_Equivalent(s, n):
    count = 0
    m = len(s)
    for i in range(n):
        rotated = s[i:] + s[:i]
        if rotated.count('1') % 2 == 1:
            count += 1
    return count
[DONE]<|endoftext|><|endoftext|><|endof
text|><|endoftext|>

Second round generation:
def odd_Equivalent(s, n):
    count = 0
    m = len(s)
    for i in range(n):
        rotated = s[i:] + s[:i]
        if rotated.count('1') % 2 == 1:
            count += 1
    return count
[DONE]<|endoftext|><|endoftext|><|endof
text|><|endoftext|><|endoftext|><|endof
text|><|endoftext|><|endoftext|><|endof
text|><|endoftext|><|endoftext|><|endof
text|><|endoftext|><|endoftext|><|endof
text|><|endoftext|><|endoftext|>......
```

(a) block generation

```
Prompt: Write a python function to find
the count of rotations of a binary
string with odd value.
[BEGIN]

First round generation:
def odd_Equivalent(s, n):
    count = 0
    m = len(s)                          ────────→ Keep
    for i in range(n):
        rotated = s[i:] + s[:i]
        if rotated.count('1') % 2 == 1:
            count += 1
    return count                        ────────→ Delete
[DONE]<|endoftext|><|endoftext|><|endof
text|><|endoftext|>

Second round generation:
def odd_Equivalent(s, n):
    count = 0
    m = len(s)                          ────────→ Generated in
    for i in range(n):                              first round
        rotated = s[i:] + s[:i]
        if rotated[-1] == '1':
            count += 1
    return count                        ────────→ Generated in
                                                   second round
# Test cases
assert odd_Equivalent("011001", 6) == 3
assert odd_Equivalent("11011", 5) == 4
assert odd_Equivalent("1010", 4) == 2
[DONE]<|endoftext|><|endoftext|><|endof
text|><|endoftext|><|endoftext|>......
```

(a) truncated block generation

Figure 4: Illustrative example comparing block generation with truncated block generation.

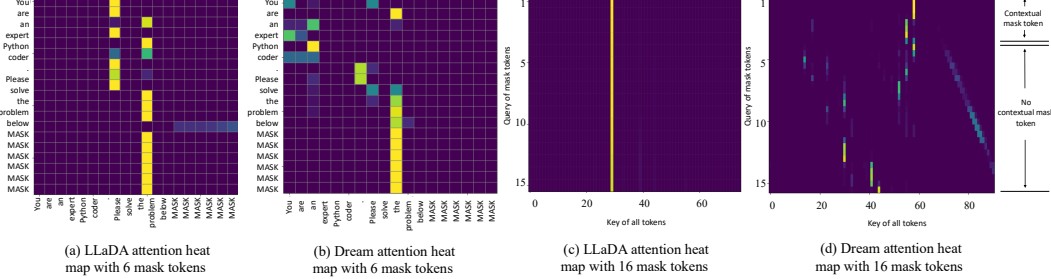

(a) LLaDA attention heat map with 6 mask tokens
(b) Dream attention heat map with 6 mask tokens
(c) LLaDA attention heat map with 16 mask tokens
(d) Dream attention heat map with 16 mask tokens

Figure 5: Attention heat maps of LLaDA and Dream.

This mechanism allows mask tokens to carry forward contextual information from the prompt. However, its application across the sequence differs between training methods.

For models trained with token-level noise (e.g., Dream), only the informative masks successfully learn this copying behavior. As shown in Figure 5 (d), non-informative masks, being too distant, fail to learn a meaningful attention pattern and instead exhibit random behavior. These semantically empty tokens are the source of attention dilution.

For models trained with sequence-level noise (e.g., LLaDA), the model is incentivized to make all mask tokens useful for the final prediction. Consequently, the copying behavior propagates throughout the entire masked sequence, as seen in Figure 5 (c). In this case, attending to any mask token is beneficial, as they all carry relevant contextual information. Therefore, models trained with sequence-level noise do not have the dilution problem.

## 4 TRUNCATED BLOCK GENERATION

In this section, we introduce the proposed method, Truncated Block Generation. We first motivate the approach by outlining the trade-offs in generative sequence length. We then detail the algorithm and discuss the importance the truncation step.

---

**Algorithm 1** Truncated Block Generation

---

**Require:** Prompt $c$, model $f_{\boldsymbol{\theta}}$, threshold $\gamma_1, \gamma_2, \cdots, \gamma_k$, block length $b_1, b_2 \cdots, b_k$, truncated length $l_1, l_2, \cdots, l_k$, sampling step $N$.

1: Set $r^1$ is a fully masked sequence of length $L$ at time step 1.
2: **for** $j \leftarrow 1$ to $k$ **do**             ▷ Generate $j$ th round.
3:     **for** $t \leftarrow 1$ **down to** $\frac{1}{N}$ **step** $\frac{1}{N}$ **do**      ▷ Iterate through all time steps.
4:        $s = t - \frac{1}{N}$          ▷ Calculate previous timestep: $s = t - 1/N$.
5:        **for** $i \leftarrow 1$ to $L$ **do**      ▷ Iterate through each token $i$ in the sequence (1 to $L$).
6:           **if** $r_t^i \neq M$ **then**        ▷ If token $i$ at timestep $t$ is not masked.
7:             $r_0^i = r_t^i, c^i = 1$      ▷ Keep the token unchanged and set confidence to 1.
8:           **else**          ▷ If token $i$ is masked.
9:             $r_0^i = \arg\max_{r_0^i} p_\theta(r_0^i | p_0, r_t)$    ▷ Predict the most likely token for this token.
10:            $c^i = p_\theta(r_0^i | p_0, r_t)_{r_0^i}$      ▷ Record the confidence score of this token.
11:           **end if**
12:        **end for**
13:        $n_{\text{un}} = [L(1-s)]$        ▷ The number of unmasked tokens is $n_{\text{un}}$ in timestep $s$
14:        **for** $i \leftarrow 1$ to $L$ **do**
15:           **if** $c^i \in \text{Lowest} - n_{\text{un}}(\{c^i\}_{i=1}^L)$ **then**     ▷ If confidence of token $i$ is low.
16:             $r_0^i = M$       ▷ Remask this token and select it for remasking.
17:           **end if**
18:        **end for**
19:        $r_s = r_0$          ▷ Update the sequence state.
20:     **end for**
21:     **if** $\text{len}(r_s) \leqslant \gamma_k$ **then**        ▷ If the valid length is less than a threshold.
22:        Break          ▷ Break the loop and output.
23:     **else**
24:        $r_0 = \text{Cat}\left(r_0[0 : l_j], [\texttt{MASK}] \times b_j\right)$     ▷ Remasking and appending mask.
25:     **end if**
26: **end for**
27: **return** $r_0$

---

As discussed in Section 3, MDMs trained with token-level noise faces the "attention dilution" challenge. Appending a long sequence of [MASK] tokens to a prompt dilutes the model's attention on the informative context, which degrades generation quality. However, a short masked sequence provides insufficient context for complex tasks, such as generating complete code blocks or executing a full chain of thought for mathematical reasoning. This limitation also leads to poor performance. This raises a key question: *How can we generate sequences with sufficient length without causing attention dilution?*

To address this challenge, we propose an iterative method called Truncated Block Generation. The core idea is to generate text in fixed-size blocks, allowing the model to extend the sequence as needed without being overwhelmed by an excessive number of [MASK] tokens at any single step. The pseudocode of the proposed algorithm is shown in Algorithm 1.

Specifically, the algorithm proceeds as follows: First, the model generates content within a masked block of a predefined length (line 2 to 20). Next, we measure the length of the valid generated output (i.e., the sequence before any <eos> token appears) and compare it against a continuation threshold, $\gamma$ (line 21). If the valid length is less than $\gamma$, we consider the generation complete. This indicates the model did not need the full block capacity to finish its response. If the valid length is greater than or equal to $\gamma$, we conclude the model needs more space. We then truncate the generated sequence to a shorter, predetermined length. This truncated output serves as the new context, to which a new block of [MASK] tokens is appended for the next round of generation.

This block-wise approach mitigates attention dilution by ensuring that only a manageable number of uninformative [MASK] tokens are present in any given generation step.

**Remarks.** The truncation step is important to the success of this method. Without it (i.e., a truncation length of zero), our algorithm would simplify to naive block generation, a strategy with

Table 1: Performance of Dream-7B and LLaDA-8B on coding benchmarks.

| Dataset | Model | Generation length | | | | | | |
|---------|-------|------|------|------|------|------|------|------|
| | | 32 | 64 | 128 | 256 | 512 | 1024 | **Ours** |
| MBPP | Dream-7B | 43.8 | 58.0 | 57.2 | 58.6 | 59.6 | 59.2 | 60.4 |
| Humaneval | | 26.8 | 43.9 | 48.7 | 48.1 | 43.9 | 43.9 | 52.4 |
| MBPP | LLaDA-8B | 22.4 | 36.2 | 37.0 | 36.6 | 36.8 | 37.4 | 37.6 |
| Humaneval | | 11.2 | 32.9 | 37.8 | 40.8 | 36.6 | 37.2 | 40.2 |

a significant flaw. Due to their bi-directional attention mechanism, MDMs are aware of sequence boundaries. As generation approaches the end of a block, the model is strongly biased toward producing an <eos> token to complete the output within the available space, as illustrated in Figure 4 (a). This premature termination prevents the model from generating content that naturally extends beyond the block boundary. In contrast, our truncation method re-masks the tokens near the end of the generated block (Figure 4 (b)). This action effectively removes the premature <eos> token and signals to the model that the sequence is incomplete (such as "#" token in GSM8K and "[DONE]" token in MBPP), prompting it to continue generating coherently into the subsequent block. This enables the flexible, arbitrary-length generation that our method is designed to achieve.

## 5 EXPERIMENTS

In this section, we show that our method can improve the sampling quality and faster generation speed for Dream. Then we conduct the ablation study and compare with block generation.

**Setup.** To show the effectiveness of the proposed truncated block generation, we test it using LLaDA-8B-Instruct (Nie et al., 2025) and Dream-7B-Instruct (Ye et al., 2025) on four datasets of math reasoning and code generation: GSM8K (Cobbe et al., 2021), MATH (Saxton & Hill, 2019), HumanEval (Chen et al., 2021), and MBPP (Austin et al., 2021b). We use 4-shot prompt for GSM8K and Math dataset, 3-shot prompt for MBPP dataset and 0-shot prompt for Humaneval dataset. All experiments are conducted on NVIDIA A6000 GPUs.

**Baseline.** We compare our method against the strongest sampling strategies of Dream and LLaDA. Specifically, LLaDA uses confidence-based remasking with semi-autoregressive decoding, while Dream adopts an entropy-based sampler. Since both approaches require a pre-defined sequence length, we directly write it as fixed length generation in the following context .

**Improved Test Accuracy.** For the baseline fixed-length generation method, we use generation length of 32, 64, 128, 256, 512 and 1024. As we can see in Table 1, on MBPP dataset, Dream achieves the best performance with a fixed generation length 512, but achieves a comparable result with fixed generation length 64. Thus, for code generation with truncated blocks, we set the block length to 64, the threshold to 55, and and the look ahead length to 30. As most of the questions can be in generated one or two blocks, we set the maximum number of generation blocks as 2.

We can see in the last column of Table 1, the proposed method achieves accuracies of 60.4 and 52.4 on MBPP and Humaneval dataset respectively. Because our method uses 2 blocks with 64 block length, compared with generation with fixed length of 128, our method attains 3.2 improvement on MBPP dataset and 3.7 on Humaneval dataset. Moreover, compare with the best result in all fixed generation length, our method also improve from 59.6 to 60.4 on MBPP and from 48.7 to 52.4 on Humaneval dataset. For math problems, as shown in Table 2, our method also improves the accuracy from 78.01 to 78.92 on GSM8K dataset and 42.80 to 43.98 on Math dataset. However, LLaDA predicts all tokens with the same penalty weight during training and do not suffer from the dilution problem as we discussed Section 3. From Table 1 and Table 2, we can see that our algorithm cannot improve LLaDA but achieve comparable performance.

**Improved Inference Speed.** We use the same setting of Humaneval in the section above. If we want to generate 128 tokens, the fixed length generation needs to append 128 tokens to the end of the prompt and forward the whole sentence. Compared with fixed length 128, we use two blocks with 64 mask tokens in each block for generation. First, if our generate finish in the first round, the forward times of the baseline will be the double because it will generate $2\times$ tokens. For example,

Table 2: Accuracy of Dream-7B and LLaDA-8B on math benchmarks.

| Model | Gen Length | GSM8K | | Math | |
|---|---|---|---|---|---|
| | | Flexible-Match | Strict-Match | Exact-Match | Math-Verify |
| Dream-7B | 512 | 75.51 | 75.43 | 42.80 | 37.38 |
| | 256 | 78.01 | 77.10 | 42.70 | 37.96 |
| | 128 | 66.64 | 58.68 | 35.48 | 35.26 |
| | 64 | 34.87 | 21.98 | 13.34 | 22.10 |
| | 32 | 6.52 | 3.03 | 00.32 | 5.02 |
| | **Ours** | 78.92 | 77.71 | 43.98 | 38.42 |
| LLaDA-8B | 512 | 80.21 | 73.2 | 26.76 | 29.94 |
| | 256 | 79.75 | 54.73 | 28.70 | 31.18 |
| | 128 | 74.22 | 59.96 | 30.86 | 29.52 |
| | 64 | 63.53 | 37.68 | 24.70 | 22.64 |
| | 32 | 21.22 | 11.75 | 4.16 | 18.8 |
| | **Ours** | 80.06 | 73.38 | 30.14 | 28.92 |

36% answers are generated only in one stage on Humaneval dataset. Moreover, longer length requires more computational time for each forward. If our generate finish in the second round, the forward time of the previous 64 tokens will be reduced because it has shorter length but the forward time of the left 64 tokens will be the same. As we can see in Table 3 and Table 4, if we use truncated block generation to generate 128 tokens, it achieves $1.9\times$ acceleration on Humaneval dataset and $1.6\times$ acceleration on MBPP dataset.

Moreover, we also show that our method also compatible with other accelerating method such as parallel decoding and block KV-Cache (Wu et al., 2025). As we can see in Table 3, the baseline Dream generates 13 tokens per second and achieves 48.7 accuracy. Compared with the baseline method, our method has $2\times$ generation speed and 52.4 accuracy, which are both higher than the baseline. Fast-dllm (Wu et al., 2025) adopt block KV-Cache and parallel decoding to accelerate the generation speed but it will degrade the performance. It achieves $3.5\times$ acceleration but the accuracy will decrease to 40.2. Our truncated block generation can further accelerate the generation of Fast-dllm and achieves higher accuracy. It achieves $4.3\times$ acceleration and have $46.3\%$ accuracy.

**Ablation on the threshold.** The hyperparameter threshold, which determines whether to continue generating the next block or not, influences the generation quality and needs to be tuned on a validation set. In this section, we also performed ablation studies of the threshold and have shown the result below. We use GSM8K dataset and exact-match the evaluation metric. The first round generation block is 256. As we can see in Figure 6, for both validation set and test dataset, the accuracy shows an upward trend with the threshold and reaches the peak 44.0 at 254. After that, the accuracy start decreasing drastically.

**Ablation on truncated length and Compare with block generation.** The truncated length also influences the quality of the sampling. When the truncated length is 0, our algorithm degrades to traditional block generation (Arriola et al., 2025). From Table 5, We can see that the performance of our truncated block generation on all the four benchmarks are all higher than the block generation. Moreover, we also test the model performance with different truncated lengths. We do experiment on Humaneval dataset and visualize the result in Figure 6 (b). We can see that in the validation set, with the increase of the truncated length, the accuracy will first increase and then decrease. And both the test set and validation set attain highest score with truncated length 30.

## 6 RELATED WORKS

**Diffusion Language Models.** Diffusion models have achieved great success in the continuous domain (Sohl-Dickstein et al., 2015; Ho et al., 2020; Karras et al., 2022). A simple approach is to map tokens into continuous embeddings and perform diffusion process in continuous space (Li et al., 2022; Han et al., 2022; Mahabadi et al., 2023). Alternatively, some methods directly train a discrete

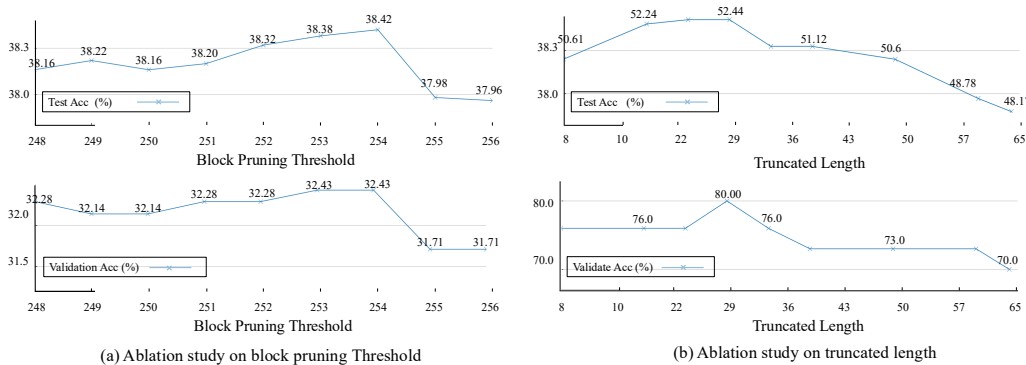

Figure 6: Ablation study on threshold and truncated length.

| Method | TPS | Acc |
|---|---|---|
| Dream-7B | 13 (1.0×) | 48.7 |
| + Ours | 25 (1.9×) | **52.4** |
| + Fast-dllm | 46 (3.5×) | 40.2 |
| + Fast + Ours | **57** (4.3×) | 46.3 |

Table 3: Comparison with Fast-dllm on Humaneval.

| Method | TPS | Acc |
|---|---|---|
| Dream-7B | 1.58 (1.0×) | 57.2 |
| + Ours | 2.48 (1.6×) | **60.4** |
| + Fast-dllm | 32.8 (20 ×) | 55.2 |
| + Fast + Ours | **42.2** (26 ×) | 55.8 |

Table 4: Comparison with Fast-dllm on MBPP.

| Dataset | Block | Ours |
|---|---|---|
| HE | 48.1 | 52.4 |
| MBPP | 58.0 | 60.4 |
| Math | 43.6 | 44.0 |
| GSM8K | 66.6 | 78.9 |

Table 5: Comparison with block generation.

diffusion models on the discrete vocabulary space (Sohl-Dickstein et al., 2015; Austin et al., 2021a; Lou et al., 2023; Nie et al., 2025; Ye et al., 2025). In this formulation, diffusion models forward steps progressively map original tokens to [MASK] tokens or random tokens, which corresponds to absorbing diffusion kernel and uniform diffusion kernel, and the reverse process reconstructs the original text from these noised sequences. Building on the above analysis, lots of works scaling the Masked diffusion models to billion-parameter scale (Nie et al., 2025; Ye et al., 2025; Khanna et al., 2025; Zhu et al., 2025). Both of them adopt the absorbing diffusion kernel, which maps original tokens to [MASK] tokens in the forward process. LLaDA series (Nie et al., 2025; Zhu et al., 2025) trained diffusion models from scratch using direct mask prediction and sentence level noise loss reweighting. Dream series (Xie et al., 2025; Ye et al., 2025) used ARMs for model initialization and trained diffusion models using shift mask prediction and token level noise loss reweighting.

**Inference Remasking strategies for dLLMs.** Diffusion large language models inference are based on low-confidence remasking (Zhu et al., 2025; Nie et al., 2025; Ye et al., 2025). Specifically, similarly to Chang et al. (2022), they remask the $\frac{t}{s}$ of predicted tokens with the lowest confidence based on the predictions, called low-confidence remasking. Moreover, Kim et al. (2025) proposed to use top probability margin remasking strategy instead of low-confidence remasking strategy, which increases the performance on several planning benchmarks.

**Block generation in dLLMs.** Block generation or semi-ar generation are widely used in currents diffusion language models (Arriola et al., 2025; Nie et al., 2025; Zhu et al., 2025; Wu et al., 2025). Arriola et al. (2025) proposed a block-wise extension of the D3PM framework (Austin et al., 2021a) to generate arbitrary-length sequences. And LLaDA (Zhu et al., 2025; Nie et al., 2025) also adopt this strategy in their models. Wu et al. (2025) introduces KV-cache in their block-wise decoding.

# 7 CONCLUSION

In this paper, we provide empirical and theoretical evidence that excessive redundant mask tokens will dilute the contextual attention of Dream model and degrade its performance. We also show that both the contextual mask tokens of Dream and all the mask tokens of LLaDA will copy a token from the prompt as query at predict the token. Inspired by the observation, we propose truncated block generation for diffusion language models sampling, which leads to faster generation speed, high quality generation, and support flexible generation. We conduct extensive experiments to visualize our observation and validate the effectiveness of the proposed algorithm.

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

## A  THE USE OF LARGE LANGUAGE MODELS

We used LLMs for grammar checking and wording improvement, ensuring it did not alter the text's meaning or add references.

## B  PROOF OF THEOREM 1

Since $\frac{w(\boldsymbol{x}_{j+1},j+1,n)}{w(\boldsymbol{x}_j,j,n)} = p$, we have $\frac{w(\boldsymbol{x}_{j+k},j+k,n)}{w(\boldsymbol{x}_j,j,n)} = p^k$. We denote $w(\boldsymbol{x}_{c+1}, c+1, n) = C$. Then, for all $\epsilon > 0$, there exists $k = \log_p \frac{\epsilon}{C}$, such that for all $u > k$, $w(\boldsymbol{x}_u, u, n) < w(\boldsymbol{x}_k, k, n) = Cp^k \leqslant \epsilon$.

## C  PROOF OF THEOREM 2

For the $X_1$ defined in Section 3, we further append $m_2 - m_1$ mask tokens to its end and denote it as $X_2$:

$$X_1 = \begin{bmatrix} P_0 \\ M_1 \end{bmatrix} \in \mathbb{R}^{1 \times (c+n_1) \times d} \quad X_2 = \begin{bmatrix} P_0 \\ M_1 \\ M_2 \end{bmatrix} \in \mathbb{R}^{1 \times (c+n_1+n_2) \times d}, \tag{8}$$

where $n_1 = m_1$ and $n_2 = m_2 - m_1$. The query of $X_1$ and $X_2$ can be calculated as:

$$Q_1 = X_1 W_Q = \begin{bmatrix} P_0 W_Q \\ M_1 W_Q \end{bmatrix} \quad Q_2 = X_2 W_Q = \begin{bmatrix} P_0 W_Q \\ M_1 W_Q \\ M_2 W_Q \end{bmatrix} \tag{9}$$

The key of $X_1$ and $X_2$ can be calculated as:

$$K_1 = X_1 W_K = \begin{bmatrix} P_0 W_K \\ M_1 W_K \end{bmatrix} \quad K_2 = X_2 W_K = \begin{bmatrix} P_0 W_K \\ M_1 W_K \\ M_2 W_K \end{bmatrix} \tag{10}$$

By multiplying the key and value, we have:

$$A^1 = \begin{bmatrix} P_0 W_Q \\ M_1 W_Q \end{bmatrix} \left( \begin{bmatrix} P_0 W_K \\ M_1 W_K \end{bmatrix} \right)^\top = \begin{bmatrix} P_0 W_Q W_K^\top P_0 & P_0 W_Q W_K^\top M_1 \\ M_1 W_Q W_K^\top P_0 & M_1 W_Q W_K^\top M_1 \end{bmatrix} \tag{11}$$

$$A^2 = \begin{bmatrix} P_0 W_Q \\ M_1 W_Q \\ M_2 W_Q \end{bmatrix} \left( \begin{bmatrix} P_0 W_K \\ M_1 W_K \\ M_2 W_K \end{bmatrix} \right)^\top = \begin{bmatrix} P_0 W_Q W_K^\top P_0 & P_0 W_Q W_K^\top M_1 & P_0 W_Q W_K^\top M_2 \\ M_1 W_Q W_K^\top P_0 & M_1 W_Q W_K^\top M_1 & M_1 W_Q W_K^\top M_2 \\ M_2 W_Q W_K^\top P_0 & M_2 W_Q W_K^\top M_2 & M_1 W_Q W_K^\top M_2 \end{bmatrix} \tag{12}$$

After softmax, we can get the attention score:

$$\text{Softmax}(A^1)_{kz} = \frac{\exp(A^1_{kc})}{\sum_{i=1}^{c+n_1} A^1_{ki}} \tag{13}$$

$$\text{Softmax}(A^2)_{kz} = \frac{\exp(A^2_{kz})}{\sum_{i=1}^{c+n_1+n_2} \exp(A^2_{ki})}, \quad \text{where } z \in [0, c], \quad k \in [c, n_1] \tag{14}$$

It is easy to show that

$$\text{Softmax}(A^2)_{kz} = \frac{\exp(A^2_{kz})}{\sum_{i=1}^{c+n_1+n_2} \exp(A^2_{ki})} \tag{15}$$

$$= \frac{\exp(A^1_{kz})}{\sum_{i=1}^{c+n_1} \exp(A^2_{ki}) + \sum_{i=c+n_1}^{n_2} \exp(A^2_{ki})} \tag{16}$$

$$= \frac{\exp(A^1_{kz})}{\sum_{i=1}^{c+n_1} \exp(A^1_{ki}) + \sum_{i=c+n_1}^{n_2} \exp(A^1_{ki})} \tag{17}$$

$$\leqslant \frac{\exp(A^1_{k1})}{\sum_{i=1}^{c+n_1} \exp(A^1_{ki})} \tag{18}$$

$$= \text{Softmax}(A^1_{kz}) \tag{19}$$

Thus, we have:

$$\text{CA}(\boldsymbol{m}_i^1) = \sum_{j=1}^{c+m_1} \frac{\exp(A_{kc}^1)}{\sum_{i=1}^{c+n_1} \exp(A_{ki}^1)} > \sum_{j=1}^{c+m_1} \frac{\exp(A_{k1}^1)}{\sum_{i=1}^{c+n_1} \exp(A_{ki}^1)} = \text{CA}(\boldsymbol{m}_i^2) \qquad (20)$$

## D  DETAILED EXPERIMENT SETTING

For Dream model, when sampling, we set dtype as "bfloat16", temperature as 0.1,top_p as 0.9 and alg as "entropy". For Fast-dllm, we set the block of KV-cache as 32.

For our truncated block generation, as shown in Table 6, for MBPP dataset, we set the block length as 64, truncated length as 32, and threshold as 55. For Humaneval dataset, we set the block length as 64, truncated length as 32, and threshold as 55. For GSM8K dataset, we set the block length as 128, truncated length as 64, and threshold as 127. For Math dataset, we set the block length as 256, truncated length as 128, and threshold as 255.

|                  | MBPP | Humaneval | GSM8K | Math |
|------------------|------|-----------|-------|------|
| Block length     | 64   | 64        | 128   | 256  |
| Truncated length | 32   | 32        | 64    | 128  |
| Threshold        | 55   | 55        | 127   | 255  |

Table 6: Detailed hyper-parameters setting.

## E  DETAILED COMPARISON OF DIFFERENT SAMPLING METHODS

In this section, we provide a detailed comparison of different methods on LLaDA using the Humaneval dataset and present the results in Table 7. We compare our approach against confidence-based remasking with and without fixed-length semi-ar generation, as well as block-wise sampling Arriola et al. (2025). With a fixed generation length of 128, LLaDA with confidence-based remasking alone achieves 8.5 accuracy, while adding semi-autoregressive sampling improves the accuracy to 37.0. Block diffusion sampling attains 17.1 accuracy. Our method using 64+64 blocks achieves 37.6 accuracy, outperforming all of the above baselines.

| Method                        | Humaneval |
|-------------------------------|-----------|
| Block diffusion               | 17.1      |
| confidence remasking w/o semi-ar | 8.5    |
| confidence remasking w/ semi-ar  | 37.0   |
| Ours                          | 37.6      |

Table 7: Detailed comparison of different sampling methods

## F  GENERAL TASK

For a general ability testing, We tested BBH (Suzgun et al., 2023) and present the results in Table 8, which consists of 23 particularly challenging BIG-Bench tasks spanning traditional NLP, mathematics, commonsense reasoning, and question answering. We can see that our method also lead to better performance.

| Method | Semi-ar + Confidence based remasking | | Ours |
|--------|------|------|-----------|
| Gen Len | 128 | 256 | 128 + 128 |
| Acc | 50.68 | 56.60 | 57.12 |

Table 8: Accuracy of LLaDA-8B on BBH benchmarks.

