# OpenReview forum: "Diffusion with Truncated Blocks: Towards Fast and High-Quality Text Generation using Truncated Block Generation"
_ICLR.cc/2026/Conference — ICLR 2026 Conference Withdrawn Submission_

### Official Review · Reviewer_Memz · 2025-10-30

**Soundness:** 3
**Presentation:** 3
**Contribution:** 3
**Rating:** 4
**Confidence:** 4

**Summary:**

The paper introduces Truncated Block Diffusion, a decoding strategy for token-level diffusion language models (DLMs) that mitigates the attention dilution problem — where excessive masked tokens reduce focus on informative context. The proposed method dynamically truncates and resumes generation in smaller blocks, allowing the model to maintain attention on semantically meaningful regions while supporting flexible-length decoding. The authors present theoretical analysis of how token-level noise leads to attention dilution and propose truncation to maintain contextual density. Experiments on code (HumanEval, MBPP) and math reasoning tasks (GSM8K, MATH) show improved accuracy and efficiency over baselines such as Dream and LLaDA, with results consistent across sequence lengths. While the technical design is well motivated and validated, the paper omits detailed analysis of failure cases or ablations regarding truncation sensitivity. The writing also tends to overemphasize strengths without clarifying limitations in robustness, calibration dependency, or computational trade-offs. Claims of compatibility with other decoding accelerations (e.g., Fast-dLLM) are promising but should be stated more cautiously since direct integration experiments are not reported.

**Strengths:**

1. The paper identifies and analyzes a concrete limitation of token-level diffusion (attention dilution), offering both theoretical reasoning and empirical validation.
2. The proposed Truncated Block Generation is conceptually simple, implementation-friendly, and does not require retraining, making it broadly applicable to existing DLMs.
3. Extensive experiments on reasoning and code tasks demonstrate consistent performance and speed improvements under fixed compute budgets.
4. The theoretical justification for truncation and the visual attention analysis are clear and insightful, helping to connect the intuition of “context density” with measured decoding quality.

**Weaknesses:**

1. The approach is primarily validated on structured domains (code and math) using Dream and LLaDA; generalization to open-ended or natural language tasks remains unclear.
2. The model’s reliance on token-level confidence for truncation may lead to instability under poorly calibrated confidence distributions.
3. The paper lacks qualitative or failure-case analysis — e.g., what happens when truncation disrupts a coherent semantic span, leading to invalid continuations.
4. No adaptive or learnable mechanism is explored for determining truncation boundaries, which could improve robustness but is omitted.
5. Broader comparisons to models with weaker calibration or longer contexts are missing, leaving uncertainty about the scalability and universality of the method.
6. The paper provides qualitative claims of acceleration but lacks comprehensive tables comparing latency and total compute under matched budgets.

**Questions:**

* **Baseline coverage and robustness under weaker confidence calibration:** Since the truncation relies on confidence-based masking, how does the approach perform on DLMs with less stable confidence scores? Would the authors include results on other diffusion LMs to verify robustness?

* **Behavior under flat or uniform confidence distributions:** When token confidences are nearly uniform, how does the model decide truncation or continuation? Can the authors provide visualization or analysis showing this failure mode?


* **Failure case analysis and adaptive truncation:** In some generations, errors may arise when truncation happens across semantically connected regions, breaking coherence. Have the authors analyzed such cases? Could an *adaptive truncation policy*—where the model learns when to truncate—alleviate this issue?

* **Compute and latency analysis:** Could the authors provide explicit comparisons of generation speed, latency, and total step counts versus fixed-length or block-decoding baselines to better quantify the claimed acceleration?

---

> ### Author Response · Authors · 2025-11-23
>
> >**W1. The approach is primarily validated on structured domains (code and math) using Dream and LLaDA; generalization to open-ended or natural language tasks remains unclear.**
>
> As suggested, we have added a new experiment evaluating LLaDA on the general-task dataset BBH (from "Challenging big-bench tasks and whether chain-of-thought can solve them", Findings of ACL 2023), which consists of 23 particularly challenging BIG-Bench tasks spanning traditional NLP, mathematics, commonsense reasoning, and question answering. We can see that our method also lead to better performance.
>
> | Method                                | Gen Len   | Acc   |
> |---------------------------------------|-----------|-------|
> | Semi-ar + Confidence based remasking | 128       | 50.68 |
> | Semi-ar + Confidence based remasking | 256       | 56.60 |
> | Ours                                  | 128 + 128 | 57.12 |
>
> >**W2 \& Q1 \& Q2. The model’s reliance on token-level confidence for truncation may lead to instability under poorly calibrated confidence distributions**
>
> We would like to clarify that our method’s truncation decision is based on the valid generation length and does not rely on token-level confidence scores. Therefore, it is not affected by potential instability arising from poorly calibrated confidence distributions.
>
> >**Q3 \& W3. The paper lacks qualitative or failure-case analysis — e.g., what happens when truncation disrupts a coherent semantic span, leading to invalid continuations.**
>
> We wish to clarify that our truncation will not lead to invalid continuations. As generation approaches the end of a block, the model is strongly biased toward finishing the generation and output tokens such as "EOS", "DONE" and "return true". The truncation is used to remask the EOS and part of the previous tokens and enable the model to produce longer and more reasonable answer. Because the model retains knowledge of the previously generated text, it can reliably regenerate the masked tokens without losing global coherence.
>
> >**W4. No adaptive or learnable mechanism is explored for determining truncation boundaries, which could improve robustness but is omitted.**
>
> Indeed, results on sensitivity to hyperparameter  have already been reported in Figure 6 of the paper.  As can be seen, the proposed method is not sensitive to the particular hyperparameter setting. For example, the accuracy fluctuates only marginally when the Truncated Length is varied from 8 to 57.
>
> It is possible that auto-tuning the  $\gamma$ and truncation length can further improve performance. However, this will be considered as future work.
>
> >**W5. Broader comparisons to models with weaker calibration or longer contexts are missing, leaving uncertainty about the scalability and universality of the method.**
>
> Current dLLMs, such as LLaDA and Dream, have not been evaluated on these tasks. We do think it is important for LLMs and will we will list it as our future work to explore.
>
> For a general ability testing, We have added BBH
> (from "Challenging big-bench tasks and whether chain-of-thought can solve them", Findings of ACL 2023),
> which consists of 23 particularly challenging BIG-Bench tasks spanning traditional NLP, mathematics, commonsense reasoning, and question answering. We can see that our method also lead to better performance.
>
> | Method                                | Gen Len   | Acc   |
> |---------------------------------------|-----------|-------|
> | Semi-ar + Confidence based remasking | 128       | 50.68 |
> | Semi-ar + Confidence based remasking | 256       | 56.60 |
> | Ours                                  | 128 + 128 | 57.12 |
>
> >**Q4 \& W6. The paper provides qualitative claims of acceleration but lacks comprehensive tables comparing latency and total compute under matched budgets.**
>
> The generation speed (tokens per second) has been reported in Tables 3 and 4 of the paper. We compare our approach with two fixed-length generation baselines: entropy-based sampling and Fast-DLLM, which is further enhanced with KV-cache and parallel sampling. Compared with the entropy-based sampler, our method achieves a 1.9× speed-up on the HumanEval dataset and a 1.6× speed-up on MBPP, while simultaneously improving generation quality. Moreover, our method also accelerates Fast-DLLM by 1.2× on HumanEval and 1.3× on MBPP, again with improved output quality.

---

> > ### Comment · Reviewer_Memz · 2025-11-28
> >
> > Thank you for the detailed response and the expanded set of experiments.  The inclusion of BBH is helpful, and I agree that large diffusion LMs currently lack established evaluations for fully open-ended generation tasks, so using BBH as a general-ability benchmark is reasonable in today’s landscape. I have only a couple of remaining points where I would appreciate further clarification:
> >
> > ### **1. Confidence dependency vs. Algorithm 1**
> >
> > In the response, the authors mention that the method does not rely on token-level confidence and is therefore not affected by confidence calibration issues. However, Algorithm 1 (lines 15–18) explicitly uses confidence values \(c_i\) to decide which tokens to remask during denoising.
> >
> > Could the authors clarify whether the *overall* decoding process still fundamentally depends on confidence calibration (even if the truncation decision itself does not), and  how stable this behavior is for diffusion LMs whose confidence distributions may be less calibrated than Dream/LLaDA? A brief explanation would help assess the robustness of the method when applied to models beyond the ones evaluated.
> >
> > ### **2. Failure cases and semantic consistency**
> >
> > The response states that truncation will not disrupt semantic continuity because the model can reliably regenerate masked segments.  However, the paper only shows successful examples, and no failure-case analysis is provided. If possible, could the authors share one or two representative failure cases, or a brief empirical observation on how often semantic inconsistency appears when truncation splits a coherent span? This would help reviewers better understand the practical behavior and limitations of the method.
> >
> > Thank you again for the thoughtful response and the additional experiments.
> > Clarifications on the above two points would help solidify the understanding of the method’s robustness and general applicability.

---

### Official Review · Reviewer_q85c · 2025-10-31

**Soundness:** 3
**Presentation:** 3
**Contribution:** 3
**Rating:** 4
**Confidence:** 4

**Summary:**

Truncated Block Generation combats attention dilution in diffusion LMs with token-level noise by decoding in short blocks and truncating before rollover, improving code/math accuracy and throughput under compute parity.

**Strengths:**

1. Formalizes “attention dilution” under token-level noise with definitions/claims plus intuitive visual evidence.
2.  Training-free; integrates with existing diffusion LMs without architecture changes.
3. Truncation removes boundary cues that trigger early <eos>, reducing “half-baked” spans.
4. Consistent gains on HumanEval/MBPP (and math sets) alongside higher tokens/sec.
5. Stacks with Fast-dllm; retains more accuracy than using acceleration alone.
6. Threshold-gated continuation makes length flexible instead of being bound to a single fixed [MASK] tail.

**Weaknesses:**

1.  Benefits concentrate on token-level noise (e.g., Dream); for sequence-level noise (e.g., LLaDA) the lift is muted—generalizability is constrained.
2. Block length, truncation length, and threshold γ all matter; best settings vary by task, raising tuning costs.
3. Focuses on code/math; lacks long-form generation, dialogue coherence, factuality, or human evals.
4. Compared to fixed-length, naive block, and Fast-dllm, but fewer head-to-heads with semi-autoregressive / multi-step remasking and other modern decoding strategies.
5. The “information value” modeling is stylized; real multi-layer attention/copy dynamics may deviate, so theory–practice gaps can appear on new models/data.

**Questions:**

1. Systematically sweep block length, truncation length, and the threshold γ; report mean ± σ across multiple random seeds and show performance/latency trade-off curves.
2. Replace raw max-softmax triggers with entropy/energy-based or calibrated-confidence signals; auto-tune γ and truncation length online to reduce hand-tuning and improve robustness OOD.
3. Evaluate beyond code/math: include long-context reasoning, dialogue coherence, and factual QA; add broader code sets (e.g., DS-1000) plus long-form generation tasks.

---

> ### Author Response · Authors · 2025-11-23
> **Part 1**
>
> >**W1. Benefits concentrate on token-level noise (e.g., Dream); for sequence-level noise (e.g.,
> LLaDA) the lift is muted—generalizability is constrained.**
>
> We would like to clarify that even under sequence-level noise, our method delivers comparable accuracy while offering substantially faster inference. For example, as shown in Table  1, our approach achieves an accuracy  of 37.6\% on MBPP using only 64+64 blocks, whereas the semi-AR + confidence-based remasking baseline requires a fixed generation length of 1024 tokens to reach similar performance level. For a detailed quantitative and qualitative analysis of the speed improvements, please refer to Tables 3 and 4 as well as the "Improved Inference Speed" section.
>
> >**W2. Block length, truncation length, and threshold $\gamma$ all matter; best settings vary by task, raising tuning costs.**
>
> As shown in Figure 6, the performance of our method is not sensitive to the  choice of hyperparameters. For example, the accuracy fluctuates only marginally when the Truncated Length is varied from 8 to 57.
>
> >**W3. Focuses on code/math; lacks long-form generation, dialogue coherence, factuality, or human evals.**
>
> Thanks for your suggestion. However, current dLLMs, such as LLaDA and Dream, have not been evaluated on these tasks.
> To address your concern, we have added a new experiment evaluating LLaDA on the general-task dataset BBH
> (from "Challenging big-bench tasks and whether chain-of-thought can solve them", Findings of ACL 2023),
> which consists of 23 particularly challenging BIG-Bench tasks spanning traditional NLP, mathematics, commonsense reasoning, and question answering. We can see that our method also lead to better performance.
>
> | Method                                | Gen Len   | Acc   |
> |---------------------------------------|-----------|-------|
> | Semi-ar + Confidence based remasking | 128       | 50.68 |
> | Semi-ar + Confidence based remasking | 256       | 56.60 |
> | Ours                                  | 128 + 128 | 57.12 |
>
> >**W4. Compared to fixed-length, naive block, and Fast-dllm, but fewer head-to-heads with semi-autoregressive / multi-step remasking and other modern decoding strategies.**
>
> We wish to clarify that the LLaDA baseline we compared is using semi-ar + confidence based remasking method, and the Dream baseline is using entropy-based sampler. These sampling methods are the best in their papers.  Because these methods also require a pre-defined fixed length, so we directly write it as fixed length generation.  We have clarified this in our revised paper.
>
> Moreover, we give a detailed comparison of different methods below.   With fix-length generation 128, LLaDA with only confidence based remasking achieves 8.5 acc, LLaDA with confidence based remasking + semi-ar sampling achieves 37.0 acc.  Our method with 64 + 64 blocks achieves 37.6 acc, which outperforms the methods above.
>
> | Method                | Humaneval |
> |-----------------------------------|-----------|
> | confidence remasking w/o semi-ar  | 8.5       |
> | confidence remasking semi-ar      | 37.0      |
> | Ours                              | 37.6      |
>
> >**W5. The “information value” modeling is stylized; real multi-layer attention/copy dynamics may deviate, so theory–practice gaps can appear on new models/data**
>
> We admit that theory–practice gaps can appear on new models/data. But we have tested
> the two mainstream dLLMs (Dream and LLaDA), and demonstrated that it consistently accelerates sampling and improves generation quality both experimentally and theoretically. We are also exploring data/model independent methods, which we will include as part of our future work.
>
> >**Q1. Systematically sweep block length, truncation length, and the threshold $\gamma$; report mean across multiple random seeds and show performance/latency trade-off curves.**
>
> The ablation study is presented in Figure 6 of the paper. Regarding the mean over multiple random seeds, we would like to clarify that all current dLLMs use a temperature of 0, so the experiments are deterministic and do not involve randomness. As for performance–latency trade-off curves, we would like to clarify that there is in fact no trade-off in our setting.
> As shown in Tables 3 and 4, for the same number of generated tokens, our method achieves higher tokens-per-second (TPS), which directly implies lower latency while simultaneously improving performance.
>
> >**Q2. Replace raw max-softmax triggers with entropy/energy-based or calibrated-confidence signals; auto-tune $\gamma$ and truncation length online to reduce hand-tuning and improve robustness OOD**
>
> For both Dream and LLaDA, we follow their original settings by adopting the confidence-based sampler and entropy-based sampler as their respective baselines and as components of our sampler. We have clarified this in the revised paper.
>
> It is possible that auto-tuning the $\gamma$ and truncation length can further improve performance. However, this will be considered as future work.

---

> ### Author Response · Authors · 2025-11-23
> **Part 2**
>
> >**Q3. Evaluate beyond code/math: include long-context reasoning, dialogue coherence, and factual QA; add broader code sets (e.g., DS-1000) plus long-form generation tasks.**
>
> Current dLLMs, such as LLaDA and Dream, have not been evaluated on these tasks.
> To address your concern, we have added a new experiment evaluating LLaDA on the general-task dataset BBH (from "Challenging big-bench tasks and whether chain-of-thought can solve them", Findings of ACL 2023), which consists of 23 particularly challenging BIG-Bench tasks spanning traditional NLP, mathematics, commonsense reasoning, and question answering. We can see that our method also lead to better performance.
> | Method                                | Gen Len   | Acc   |
> |---------------------------------------|-----------|-------|
> | Semi-ar + Confidence based remasking | 128       | 50.68 |
> | Semi-ar + Confidence based remasking | 256       | 56.60 |
> | Ours                                  | 128 + 128 | 57.12 |

---

### Official Review · Reviewer_EtJq · 2025-11-01

**Soundness:** 2
**Presentation:** 2
**Contribution:** 1
**Rating:** 0
**Confidence:** 5

**Summary:**

The paper proposes Truncated Block Generation (TBG), a decoding method for diffusion-based LLMs trained with token-level noise (e.g., Dream).
It argues that such models suffer from attention dilution-where long tails of uninformative mask tokens weaken attention focus.
TBG generates text in smaller masked blocks and truncates partial outputs iteratively, claiming to improve efficiency and text quality for long sequences.

However, the approach closely overlaps with semi-autoregressive or blockwise diffusion decoding already established in SSD-LM and Block Diffusion (BD3-LMs). The only new element-truncation-is a heuristic, not a fundamental algorithmic advance. Baselines are incomplete, and empirical support for both speed and “attention dilution” is weak.

**Strengths:**

Identifies a plausible failure mode (attention dilution) in token-level noise training.
Simple heuristic (TBG) that is easy to implement on top of existing diffusion LLMs. However, the approach is not properly compared agianst prior baselines!
Empirical results show modest improvements on some reasoning and coding benchmarks, still not compared to prior methods!

**Weaknesses:**

* The paper lack novelty and fails to compare against prior blockwise diffusion baselines such as SSD-LM and BD3-LMs (https://arxiv.org/pdf/2503.09573) in comparisons, which severely undermines the claim of novelty and contribution.

The proposed Truncated Block Generation (TBG) is conceptually almost identical to SSD-LM (Han et al., 2023) and Block Diffusion / BD3-LMs (Arriola et al., 2025), which already generate text in sequential diffusion blocks conditioned on previous outputs. However, such strong
Both prior works support variable-length text generation, KV caching, and efficient blockwise denoising - the same core benefits claimed here. The only new element is the truncation heuristic, where the generated block is shortened before continuation. However, Looking into appendix this approach comes with heavily tuning the hyper-parameters which questions the practicality of this approach in realworld setting.


The main baseline is “Dream with full-length mask decoding,” which is known to perform poorly on long outputs and serves as a weak strawman.



* Unconvincing theoretical framing (“attention dilution”)
The “attention dilution” argument - that many uninformative MASK tokens from token-level noise distract the attention distribution - is intuitively reasonable but not experimentally validated.


* The analysis merely restates the known property that softmax weights are normalized over all keys; it does not establish a causal connection between dilution and degraded text quality.

The authors argue that truncating uninformative MASK positions (or reducing context length)  provide improvements.
Since sequence-level noise models (like LLaDA) are unaffected by this problem, a simpler alternative would be to adjust training rather than add decoding heuristics.

* Strong baselines such as SSD-LM, Block Diffusion (BD3-LMs), and LLaDA with standard decoding are not compared, even though they directly address the same limitations. without proper comparison, it is unclear if the approach bring any benefits. It remains unclear whether TBG helps other diffusion LMs, semi-autoregressive LMs, or sequence-level-noise systems that already avoid dilution.


No comparison with autoregressive models on runtime or accuracy, despite the claim of “fast and high-quality generation.”

“Faster inference” is repeatedly claimed but not substantiated: the paper reports no wall-clock time, throughput (tokens/sec), FLOPs, or NFEs (number of function evaluations).

TBG introduces multiple iterative decoding rounds (generate -> truncate -> repeat), each requiring diffusion denoising, which likely increases latency. this is good to clarify this in the paper.


* High hyperparameter sensitivity and tuning overhead

TBG depends on several heuristic hyperparameters (block length, truncation length, threshold), which are tuned per dataset . Looking into appendix it looks like they are heavily tuned. this would question practicality of this method.  In contrast, block diffusion and SSD-LM decoding work robustly across datasets without such per-task adjustments. This extensive tuning contradicts the claim of being a “simple, fast decoding algorithm.”

There is no direct causal experiment showing that truncation specifically restores attention concentration or improves generation quality for the same noise schedule.

**Questions:**

How is TBG different from SSD-LM or Block Diffusion decoding beyond the truncation heuristic?

Paper needs to compare with SSD-LM and BD3-LM as prior baselines. Could you provide the comparisons?

If attention dilution only arises in token-level noise (Dream), why not just adopt sequence-level noise (LLaDA)?

could you provide wall-clock or NFE comparisons supporting “faster inference”?

How sensitive is TBG to hyperparameter tuning (block length, truncation, threshold)?

---

> ### Author Response · Authors · 2025-11-23
> **Part 1**
>
> Dear Reviewer EtJq,
>
> We noticed that you rewrote your entire review on November 20, which is eight days after the rebuttal period began.
>
> The following is our rebuttal to your original review. We will provide a separate rebuttal addressing your revised review as soon as possible.
>
> >**W1 \& Q1 \& Q2  (1) The paper lack novelty.  (2) Compared with Block Diffusion / BD3-LMs, the only new element is the truncation heuristic. (3) The paper fails to compare against prior blockwise diffusion baselines such as SSD-LM and BD3-LMs**
>
> 1. The novelty of our method lies in providing both empirical and theoretical evidence that it effectively mitigates the dilution problem in dLLMs, improves sampling speed, and supports flexible generation. None of these are achieved by existing sampling methods.
>
> 2. Differences between the proposed method and blockwise diffusion such as BD3-LMs:
> (i) Blockwise diffusion models such as BD3-LMs use **block-causal attention**; while our method uses **full bi-directional attention**, which is designed for models trained by full bi-directional attention such as LLaDA and Dream.  The block-causal attention matrix is a block-wise lower triangular matrix, whose generation method is more similar to ARM (i.e., endding the generation with an EOS token) but not dLLM (which generates a fixed-length paragraph). Directly applying block diffusion sampling
> to dLLMs lead to very poor performance. Please see the comparison below. (ii) Block-wise diffusion models terminate generation as soon as an EOS token appears. Instead, our method determines when to stop based on the valid generation length of each block. If the valid generation length exceeds a predefined threshold, we remask the EOS token along with part of the preceding tokens even if an EOS is generated. This mechanism not only yields better performance but also enables flexible control over the final generation length.
>
> 3. For SSD-LM and BD3-LMs, we would like to clarify that they are unconditional diffusion language models and therefore do not support conditional inputs. For example, SSD-LM is a continuous diffusion language model that always begins generation from Gaussian noise, making it unsuitable for real-world conditional tasks. As shown in their paper, all results are reported in terms of perplexity rather than accuracy. To address your concern, we have added a comparison using the block-diffusion sampling method on LLaDA. As shown in the table below, its accuracy is significantly lower than that of our method.
>
> | Method          | Humaneval  Acc| MBPP Acc|
> |-----------------|-----------|------|
> | Block diffusion | 17.1      | 21.2 |
> | Ours            | 40.2      | 37.6 |
>
> >**W2. The main baseline is “Dream with full-length mask decoding,” which is known to perform poorly on long outputs and serves as a weak strawman.**
>
> In fact, for both Dream and LLaDA, all their existing sampling strategies are based on full-length mask decoding. Although Dream adopts entropy-based sampling and LLaDA uses semi-AR decoding with confidence-based remasking, both methods still require a predefined generation length.
>
> >**W3. Unconvincing theoretical framing (“attention dilution”) The “attention dilution” argument - that many uninformative MASK tokens from token-level noise distract the attention distribution - is intuitively reasonable but not experimentally validated.**
>
> We would like to clarify that this “attention dilution” phenomenon has indeed been empirically validated in our paper.
> As stated in line 193-201: We empirically validate this phenomenon in Figure 3. As we can see in Figure 3 (a) and Figure 3 (b), given a prompt ``You are a expert python coder”, when the number of appended MASKs is 64, the
> attention of the first mask token to the [You] token is $2.1\times 10^{-5}$. However, if the number of appended
> MASKs is 1024, the attention of the first mask token to the [You] token will decrease to $5.3\times10^{-6}$.
> The same phenomenon can be observed for attention on all of the prompt tokens. Moreover, we also visualize how the contextual attention affected by the number of mask tokens and show the result in Figure 3 (c). We draw the curve of three samples and the mean on the MBPP dataset, and we can
> see that the contextual attention is decreasing with the increase of the number of mask tokens. This
> demonstrates that an excess of masks diverts focus from the crucial context provided by the prompt.

---

> ### Author Response · Authors · 2025-11-23
> **Part 2**
>
> >**W4. The analysis merely restates
> the known property that softmax weights are normalized over all keys; it does not establish a causal connection between dilution and degraded text quality.**
>
> 1. First, we would like to clarify that softmax is not normalized over all keys but instead normalized over each row of the attention matrix. Second, what we want to state is that excessive redundant mask tokens will dilute the contextual attention and degrade its performance, which as far as we know has not been pointed out in existing studies.
>
> 2. In the following, we illustrate the causal connection between dilution and degraded quality by retrieval signal-to-noise ratio (SNR), which measures the proportion of noise relative to the signal in a retrieval task [Xu et al., Understanding transformer from the perspective of associative memory, NeurIPS-25].
> Specifically considering $q_t = k_j$, retrieving any $v_j$ is performed as:
>
> $$
> \begin{aligned}
> o_t = \sum_{i=1}^L \exp(k_j^\top k_i) v_j =  \sum_{i=1}^L ( v_i\phi(k_i)^\top ) \phi(k_j)
> = v_j \phi(k_j)^\top \phi(k_j) + \sum_{i\neq j} ( v_i\phi(k_i)^\top ) \phi(k_j),
> \end{aligned}
> $$
>
> where $\phi$ is the feature mapping corresponding to the exp kernel. The SNR is defined as
> \begin{align*}
>     \text{SNR} = \frac{v_j \phi(k_j)^\top \phi(k_j)}{\sum_{i\neq j}   ( v_i\phi(k_i)^\top  )  \phi(k_j)}.
> \end{align*}
> With more MASK tokens appended, the numerator progressively decreases, whereas the denominator keeps increasing, which leads to decreased signal-to-noise ratio and degraded performance.
>
> >**Q3 \& W5. The authors argue that truncating uninformative MASK positions (or reducing context length) provide improvements. Since sequence-level noise models (like LLaDA) are unaffected by this problem, a simpler alternative would be to adjust training rather than add decoding heuristics.**
>
> In Dream, the authors explicitly note that sentence-level training leads to suboptimal learning, particularly for tokens with varying degrees of contextual informativeness.  Therefore, simply adjusting the training procedure cannot address the issue.
> Moreover, for LLaDA, we would like to clarify that (1) it is not unaffected by this problem; rather, it only alleviates it. Achieving the same accuracy still requires substantially longer generated sequences, which in turn increases inference time. For example, as shown in Table 1, our method achieves  an accuracy of  37.6\%  on the MBPP dataset with $64+64$ blocks. whereas the fixed generation with semi-autoregressive + confidence-based remasking approach requires a fixed length of $1024$ to achieve that accuracy. (2) Our method also accelerates sampling (see our "Improved Inference Speed" section for details) (3) In addition, our approach supports flexible-length sequence generation, while existing methods such as LLaDA cannot.
>
> >**Q4. could you provide wall-clock or NFE comparisons supporting “faster inference”?**
>
> The generation speed (tokens per second) has been reported in Tables 3 and 4 of the paper. We compare our approach with two fixed-length generation baselines: entropy-based sampling and Fast-DLLM, which is further enhanced with KV-cache and parallel sampling. Compared with the entropy-based sampler, our method achieves a 1.9× speed-up on the HumanEval dataset and a 1.6× speed-up on MBPP, while simultaneously improving generation quality. Moreover, our method also accelerates Fast-DLLM by 1.2× on HumanEval and 1.3× on MBPP, again with improved output quality.
>
>
> >**Q5. How sensitive is TBG to hyperparameter tuning (block length, truncation, threshold)?**
>
> Indeed, results on sensitivity to hyperparameter have already been reported in Figure 6 of the paper. As can be seen, the proposed method is not sensitive to the particular hyperparameter setting. For example, the accuracy fluctuates only marginally when the Truncated Length is varied from 8 to 57.

---

### Official Review · Reviewer_niRg · 2025-11-03

**Soundness:** 3
**Presentation:** 3
**Contribution:** 3
**Rating:** 6
**Confidence:** 3

**Summary:**

The paper identifies and analyzes an “attention dilution” problem that arises in masked/diffusion LLMs under token-level noising, and proposes a simple yet effective sampling strategy called Truncated Block Generation to mitigate this issue—thereby improving generation quality and speeding up inference.

**Strengths:**

1. It pinpoints and rigorously analyzes a previously overlooked “attention dilution” problem in token-level noising dLLMs, supporting the claim with both theoretical arguments and attention visualizations.

2. It proposes a simple, practical sampling algorithm—Truncated Block Generation—that directly mitigates the dilution issue by generating in short blocks and truncating to keep context informative, making the method easy to integrate into existing dLLM pipelines.

3. The approach is empirically validated: experiments and ablations show consistent quality improvements and inference speedups on code and math benchmarks (e.g., MBPP, HumanEval, GSM8K), and the paper demonstrates robustness to key hyperparameters like truncation length and threshold.

**Weaknesses:**

1. There needs to be some baseline comparison, such as adding comparisons using methods like remasking during the inference stage. Currently, there are almost no baselines.

2. I want to see how this method performs on some general tasks such as MMLU and GPQA.

**Questions:**

1. How does truncation interact with long-context reasoning or compositional generation—does repeatedly truncating and regenerating blocks risk losing global coherence or factual consistency over long outputs?

2. I observed that the longer the generated length, the better the performance. What if the generated length is 8k or even 16k? What are the advantages of this method?

---

> ### Author Response · Authors · 2025-11-23
>
> >**W1. There needs to be some baseline comparison, such as adding comparisons using methods like remasking during the inference stage. Currently, there are almost no baselines.**
>
> We would like to clarify that in the comparisons reported in Tables 1 and 2 of the main paper, we have already used each method’s strongest remasking strategy—specifically, the confidence-based remasking + semi-ar in LLaDA and the entropy-based sampler in Dream.
> We have clarified this in the revised version of the paper.
>
> >**W2. I want to see how this method performs on some general tasks such as MMLU and GPQA.**
>
> Our method is designed for reasoning tasks. On the other hand, MMLU and GPQA are multiple-choice benchmark the evaluation code does not require a predefined generation length. Since only a single token needs to be generated for each question, there is no need for any sampling strategy, and therefore our method does not provide a meaningful comparison on these two benchmarks.
>
> To address your concern, we have added a new experiment evaluating LLaDA on the general-task dataset  BBH
> (from "Challenging big-bench tasks and whether chain-of-thought can solve them", Findings of ACL 2023), which consists of 23 particularly challenging BIG-Bench tasks spanning traditional NLP, mathematics, commonsense reasoning, and question answering.
> We can see that our method also lead to better performance.
>
> | Method                                | Gen Len   | Acc   |
> |---------------------------------------|-----------|-------|
> | Semi-ar + Confidence based remasking | 128       | 50.68 |
> | Semi-ar + Confidence based remasking | 256       | 56.60 |
> | Ours                                  | 128 + 128 | 57.12 |
>
> >**Q1. How does truncation interact with long-context reasoning or compositional generation—does repeatedly truncating and regenerating blocks risk losing global coherence or factual consistency over long outputs?**
>
> Truncation will not lose global coherence or factual consistency.  As generation approaches the end of a block, the model is strongly biased toward finishing the generation and output tokens such as "EOS", "DONE" and "return true". The truncation is used to remask the EOS and part of the previous tokens and enable the model to produce longer and more reasonable answer. Because the model retains knowledge of the previously generated text, it can reliably regenerate the masked tokens without losing global coherence.
>
> >**Q2. I observed that the longer the generated length, the better the performance. What if the generated length is 8k or even 16k? What are the advantages of this method?**
>
> 1. We want to clarify that it is not always true that the longer is the generated length, the better the performance. For example, as shown in Table 1, the best performance of Dream and LLaDA on HumanEval is achieved at lengths 128 and 256, respectively, which is much higher than the cases with 512 and 1024 mask tokens. This is also consistent with the results reported in
> the  LLaDA paper (https://arxiv.org/pdf/2502.09992v1),  in which its Table 8 shows that the best performance is always achieved with length 128-512.
>
> 2. The advantage of the proposed method is that we can have flexible generation length and address the ``attention dilution'' problem when the mask sequence is long.

---

### Comment · Area_Chair_hZ2i · 2025-11-28

Dear Reviewers,

The authors have responded to your reviews. Please engage in the discussion and evaluate the authors’ rebuttal to check whether your comments have been adequately addressed, and determine whether you would like to adjust your evaluations.

Best,

Your AC

---

### Note · Authors · 2025-12-25

I have read and agree with the venue's withdrawal policy on behalf of myself and my co-authors.